# Visuomotor adaptation across the lifespan

**Holly A. Clayton** [1,2¤]*, **Sahir Abbas**[1], **Bernard Marius 't Hart** [2], **Denise Y. P. Henriques**[1,2,3]

**1** Department of Psychology, York University, Toronto, Ontario, Canada, **2** Centre for Vision Research, York University, Toronto, Ontario, Canada, **3** School of Kinesiology and Health Sciences, York University, Toronto, Ontario, Canada

¤ Current address: School of English and Liberal Studies, Seneca Polytechnic, King City, Ontario, Canada
* hollyaclayton@gmail.com

## Abstract

Being able to adapt our movements to changing circumstances allows people to maintain performance across a wide range of tasks throughout life, but it is unclear whether visuomotor learning abilities are fully developed in young children and, if so, whether they remain stable in the elderly. There is limited evidence of changes in motor adaptation ability throughout life, and the findings are inconsistent. Therefore, our goal was to compare visuomotor learning abilities throughout the lifespan. We used a shorter, gamified experimental task and collected data from participants in 5 age groups. Young children (M = 7 years), older children (M = 11 years), young adults (M = 20 years), adults (M = 40 years) and older adults (M = 67 years) adapted to a 45° visuomotor rotation in a centre-out reaching task. Across measures of rate of adaptation, extent of learning, rate of unlearning, generalization, and savings, we found that all groups performed similarly. That is, at least for short bouts of gamified learning, children and older adults perform just as well as young adults.

## Introduction

Being able to adapt our motor repertoire to changes in the environment can easily be taken for granted. Visuomotor learning allows us to become self-sufficient adults; without it we might struggle with the simplest daily tasks, such as feeding ourselves or brushing our teeth. In the laboratory, motor learning can be examined via visuomotor adaptation paradigms, where participants adapt their movements to place a misaligned cursor on a target. The most common perturbation is a visuomotor rotation, where the cursor movement is deviated from the expected direction, relative to the home position, usually by 30 or 45°. To move the cursor directly to the target, the required movement must compensate for this rotation, to which young adults usually adapt quickly [1]. While visuomotor adaptation has been extensively studied in young adults, there are far fewer studies that have examined visuomotor adaptation in children, or older adults, and many of the findings are inconsistent across experiments.

Along with compensating for cursor misalignment, adaptation leads to robust changes in performance. First, people usually continue to make "compensatory movements" even when the perturbation is no longer present. These persistent deviations are known as aftereffects and they tend to be no larger than 15°, regardless of the size of the abruptly introduced rotation [2]

**Data Availability Statement:** All data is available on OSF (https://osf.io/kgt2z/).

**Funding:** The study was funded by a Natural Sciences and Engineering Counsil of Canada (NSERC) grant to DYPH. The funders had no role in study design, data collection and

analysis, decision to publish, or preparation of the manuscript.

**Competing interests:** The authors have declared that no competing interests exist.

but do not last long, decaying rapidly as participants revert to baseline performance [3]. Adaptation is also not just restricted to the trained location but can generalize to novel target locations that are close in proximity to the trained target [4, 5]. When exposed to the same perturbation after a period of time, or after unlearning, adaptation during the second exposure is faster; this faster re-learning is known as savings [6].

Studies involving children have indicated that they may have difficulty adapting to cursor rotations. Some studies show that younger children do not compensate for cursor rotations as much as older children [7, 8], or young adults [9] while others find no differences [10]. One study found that children exhibited a slower rate of early adaptation, but not late adaptation, when compared to adults [11]. Similar mixed results have been found regarding whether young children produced reach aftereffects to the same extent as older children and adults. One study found smaller aftereffects for children younger than 8 [7] while other studies found no differences [8, 10]. Slower de-adaptation rates have also been found in children [11]. It is not known whether children show similar generalization patterns as adults. The only study that explored savings in children did not find evidence of a quicker relearning rate when children re-adapted to the same 60˚ cursor rotation they encountered 10 hours earlier [12], although there was no adult control group to compare the children's performance to. Therefore, it is unclear whether children, especially those younger than 8, possess fully mature visuomotor learning abilities, like those of young adults.

Studies involving older adults have also indicated that, like children, they may have difficulty adapting to cursor rotations. Some studies found no differences in compensation for a rotated cursor between older and younger adults [13–15], while others found that older adults adapted at a slower rate [11, 16], or to a lesser extent [13, 17–21], than young adults. Differences in compensation tend to emerge when training with larger cursor rotations and are attributed to deficits in cognitive components of learning in older adults [13, 22]. Even when their adaptation performance differed from young adults, older adults still show comparable aftereffects to young adults [13, 15], which usually decay at a similar rate [14, 21], although one study found slower de-adaptation in older adults [11]. Of these studies, only one measured generalization patterns to nearby, untrained target locations; Heuer and Hegele (2008) found older adults showed similar generalization patterns, that were of a similar magnitude, as young adults. A final aspect that has been hardly explored in older adults is savings, with a few studies [17, 22, 23] finding they exhibited savings, of a similar magnitude as young adults, and only one finding greater savings in older adults [24]. Thus, it is unclear whether the visuomotor learning capabilities of younger adults persist into old age.

The main goal of this study was to compare various properties of visuomotor learning performance across the lifespan. Given mixed evidence regarding visuomotor learning abilities in older and younger groups, we created a short, gamified experiment, that could be utilized in a familiar environment (such as at a library or camp), to better engage participants of any age. Despite the study being short, we tried to capture a variety of adaptation performance measures, such as rate of learning, generalization, and savings, which have been largely unexplored in children or older adults. Furthermore, there are hardly any studies that have compared visuomotor adaptation abilities of all age groups within the same paradigm, like we did here. We measured the rate, and extent, of visuomotor adaptation when a 45˚ counterclockwise (CCW) cursor was abruptly introduced. To do this, we first had to determine levels of baseline accuracy, and variability, when reaching with an aligned cursor, so that we could see how performance changed across adaptation training. Most studies focus on the overall magnitude of adaptation, rather than its speed, so we compared changes across blocks of 3 trials to better capture potential group differences in learning rate. We measured aftereffects, after the cursor was realigned with hand movement, to compare the sizes of aftereffects, to compare the rate of

unlearning, and to examine generalization of learning to untrained targets. We also examined the rate of re-learning, when the cursor rotation was introduced a second time, and compared levels of savings, all of which we compared across the age groups. We hypothesized that the gamified nature of the short task would prevent boredom, and fatigue, in our participants, revealing that visuomotor adaptation performance is no different across the lifespan, at least in short, gamified situations.

## Methods

### Participants

One hundred and thirty-one participants voluntarily took part in the in-person experiment outlined below. There were eighteen young children (mean age 7 years, range 5–8, 9 females, 1 left-handed), forty older children (mean age 11 years, range 9–13, 15 females, 2 left-handed), twenty young adults (mean age 20 years, range 19–22, 10 females, 1 left-handed), thirty-six adults (mean age 40 years, range 27–51, 28 females, 4 left-handed, 1 ambidextrous) and seventeen older adults (mean age 67 years, range 59–78, 11 females, 2 left-handed). All participants had corrected-to-normal vision, were pre-screened for neurological dysfunction, and most were right-handed. As the experiment was done at the public locations described below, we did not strictly control sample size and used all data that was collected.

All young children, and some older children, were recruited and tested at the Innisfil Public Library (starting July 3, 2019, and ending August 28, 2019). The remaining older children were recruited and tested at the York University Science Camp program (starting March 11, 2019, and ending July 26th, 2019). Each child's guardian indicated the child's experience using a computer mouse (not at all, very little, some, a lot, or every day). Adults of all ages were recruited through the York community lecture series event, while some were recruited through word of mouth. Some young adults were recruited from the Undergraduate Research Participant Pool at York University (and given course credit for their participation). Adult data collection began on May 1, 2019, and ended on February 28, 2020. Adults only indicated whether they use a mouse when using a computer. No monetary rewards were provided to any individual for their participation in the study. All participants, or their guardians, provided written informed-consent, and all minors provided an additional verbal assent, to participate in the study. The study was approved by the Office of Research Ethics at York University (approval #e2019-150) and the study was conducted in accordance with the ethical guidelines set by the Human Participants Review Sub-committee.

### General experimental setup

Participants sat on a chair in front of a 14" Dell laptop and held a wired computer mouse with their right hand. A miniature mouse was used when testing the children, to accommodate their smaller hands. The chair was positioned so that they could comfortably see images, as they appeared on the laptop, and reach with their right hand, represented by the mouse's cursor, from a starting home position on the screen to one of three possible target locations (Fig 1). The experiment was designed to resemble a children's game (about 10 minutes long), to help motivate children's participation, in which the targets were images of planets, while the cursor was an image of an alien, both about 1 cm wide. The home position was a 0.5 cm wide circle located at the bottom-centre of the screen (filled circle in Fig 1). The targets were displayed at approximately 10 cm distance from the home position at 45˚, 90˚ & 135˚ angles. Target and cursor images were randomized across trials from a set of 30 different planets and 45 different aliens. The programming language Python was used to display cursor movements/target positions and measure cursor movements, at roughly 60Hz.

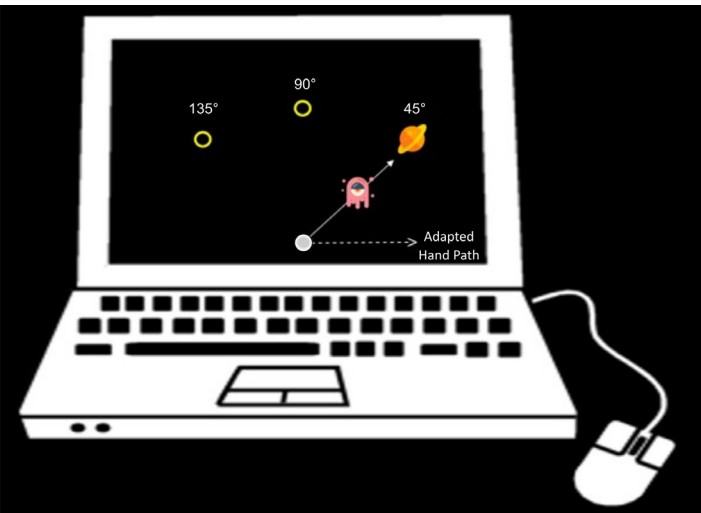

**Fig 1. Experimental setup.** Participants used the mouse to place the cursor (alien) on the target (planet). The open circles represent additional target locations that were part of the baseline and washout phases. The filled circle represents the home position.

## Procedure

The experiment comprised of 4 phases: baseline, training, washout, and relearning.

All participants completed the four phases, in the described order, during one continuous session. Participants were instructed to move the alien (mouse cursor) directly from a common home position, with their right hand, as smoothly and as accurately as possible towards the planet (target). A reach trial was complete when the centre of the mouse cursor overlapped with the target. Upon completion of the reach, the target vanished, and participants were required to return the mouse cursor to the home position before a new target appeared.

Reaches were made with either an aligned cursor or a rotated cursor. When reaching with the aligned cursor, the mouse-cursor movement was aligned with the movement of the participant's hand. During the rotated cursor blocks, the movement of the mouse-cursor was abruptly rotated 45° CCW relative to the home position. To compensate for this rotation, a participant would have to reach 45° clockwise (CW) to acquire the target with the cursor (dashed arrow in Fig 1). Participants were not informed of the cursor rotation at any point throughout the experiment.

The initial baseline phase consisted of 15 reach trials with the aligned cursor. During this phase, all 3 of the target locations (45°, 90° and 135°) were used, with 5 reaches per location. Each of the 3 locations were presented in a pseudorandom order. During the training phase the cursor was abruptly rotated 45° CCW, and this phase consisted of 45 reaches towards only the 45° target location. The washout phase was a repeat of the baseline phase, wherein participants again reached 15 times with the aligned cursor towards all 3 of the targets. Finally, during the relearning phase, participants reached 15 times with the 45° abruptly rotated cursor towards the target located at 45°. There were no time limits imposed on any task, but the experiment was designed to take only 10–15 min to keep children, and members of the public, engaged.

## Data analysis

The main goal of this experiment was to determine the effect of age on visuomotor learning processes. First, we tested if there were any differences in baseline motor performance. Then,

we explored if there were any differences in visuomotor learning abilities across various phases of the experiment. Our measurement unit was "hand angle", which is the angular difference between the hand movement path, at 1/3 of the total distance to the target, and a straight trajectory towards the target. We selected this distance to capture ballistic movements to the target, rather than corrective movements. Errors refer to the angular discrepancy between this hand angle and target angle.

Outliers were removed via pre-processing scripts in Python, when errors were larger than ±30˚ for the baseline and ±60˚ for the washout phase. For training and relearning phases, the window was asymmetrical, to incorporate the rotation, spanning from -30˚ to +75˚. Errors of this size would imply the participant did not attempt to reach directly towards the target as instructed. One middle-aged participant was removed from the data set for not showing any evidence of learning throughout the experiment.

All analyses were done in JASP version 0.16.3 (JASP Team, 2022). We used the repeated-measures function to conduct mixed ANOVAS in JASP and we defined within-subjects/between-subjects factors, as described below. For all statistical tests the alpha level was set to 0.05 and Greenhouse-Geisser corrections were applied when appropriate. Bayesian statistics are also reported with $BF_{10}$, which estimates the likelihood of the alternative hypothesis being true given our data, for each corresponding frequentist test. However, post-hoc tests were only conducted with frequentist analyses. A summary of the measures derived from each task, and the analyses conducted, are described further below. All data is available on OSF (https://osf.io/kgt2z/).

**Baseline motor control.** First, we analysed group differences in hand-cursor movements during the baseline phase. We calculated means, and standard deviations, for each participant, based on the last 9 trials in the phase (trials 7–15). These later trials are a better measure of baseline performance because they do not include trials where participants are still familiarizing themselves with the task, yet they still capture each target location exactly 3 times. Then we compared mean angular errors (to measure accuracy, or bias) and mean standard deviations of the reaching errors (to measure variability, or spread relative to the accuracy), across the five groups using two separate 1-way ANOVAs. For all further analyses described below, we subtracted the mean angular errors from the reaching blocks (average of three consecutive trials) of interest and refer to them as "reach deviations". We only analyzed variability in the baseline phase because this was the only phase where it made sense to do so. The mean standard deviation would not be a reliable measure in other phases, since the mean angular errors changed as participants adapted, de-adapted and re-adapted.

We also investigated the impact of mouse experience on children's baseline motor control. We converted the guardian's ratings of children's mouse experience into a Likert scale (not at all, 0; very little, 1; some, 2; a lot, 3; or everyday, 4). Then, we compared mean mouse experience across the two groups of children (young vs older) using a Welch T-test. Lastly, we examined whether the standard deviations of the reaching errors (our measure of variability) were related to mouse experience in children using a Pearson correlation.

**Learning rate.** To answer our main questions, we confirmed that all groups learned to counter the cursor rotation by the end of the 45 training trials. We calculated average reach deviations for block 1, block 2, block 5 and block 15 (trials 16–18, 19–21, 28–30, and 58–60, respectively). Then, we compared these means using a 4 X 5 mixed ANOVA, with block as the within-subjects factor and group as the between-subjects factor, to determine if the rate and extent of learning varied across age groups. We have also provided a more comprehensive measure of learning rate, calculated by fitting an exponential function to our data, in a supplemental file for interested readers (see S1 File).

**Aftereffects, unlearning rate, and generalization.** To confirm that each group showed aftereffects, we performed 5 separate single-sample t-tests (one for each group), to determine whether reach deviation during the first block of washout was significantly different from 0. Then we conducted a 1-way ANOVA to detect any group differences in aftereffects. Since this measure is based on a set of 3 trials, interested readers can refer to Table 1 in S1 File for an estimate of aftereffects when participants reached to the trained target (45˚) for the first time. Next, we compared group performance in block 1, block 2 and block 5 of the washout phase (trials 61–63, 64–66 and 73–75, respectively). We conducted a 3 X 5 mixed ANOVA, with block as the within-subjects factor and group as the between-subjects factor, to determine if there were group differences in how learning decayed across blocks. An estimated rate of this decay is also available in S1 File. We also wanted to confirm that all groups returned to baseline performance by the end of the 15 washout trials. We also performed 5 separate single-sample t-tests to confirm that all groups returned to baseline performance by the end of the 15 washout trials (by determining if reach deviations in the last block of washout were significantly different from 0, for each of the 5 groups). Then we compared group performance in this last block of washout using a 1-way ANOVA.

Lastly, to determine if age influenced how learning generalizes from the trained target (45˚) to untrained targets (90˚ and 135˚) during the washout phase, we calculated average reach deviations for each of the target locations, collapsing across all trials in the washout phase. We conducted a 3 X 5 mixed ANOVA, with target as the within-subjects factor and group as the between-subjects factor, to explore group differences in generalization patterns.

**Relearning rate and savings.** Finally, we compared group differences in the rate of re-learning, and savings, during the last phase of the experiment. We calculated reach deviations, for each group, for block 1, block 2 and block 5 of re-learning (trials 76–79, 79–81, and 88–90, respectively). Then we conducted a 3 X 5 mixed ANOVA, with block as the within-subjects factor and group as the between-subjects factor, to explore group differences in re-learning rates when we introduced the cursor rotation a second time. An estimated rate of re-learning is available in S1 File.

To assess group differences in savings, we compared the percentage of the final learning extent attained in block 1 of training (dividing each participant's reach deviation in block 1 by the average total learning extent achieved by all groups in block 15, which was 41.68˚) with the percentage of learning extent attained in block 1 of relearning (using the same calculation). Since none of the groups returned to baseline by the end of the washout phase, we adjusted their reach deviations in block 1 of relearning to account for this [25]. Instead of subtracting their original baseline average, we subtracted the average reach deviation from block 5 of washout and recalculated the percentages of learning extent in block 1 of re-learning. We conducted a 2 X 5 mixed ANOVA, with phase (percentage of learning extent achieved in block 1 of training or relearning) as the within-subjects factor and group as the between-subjects factor, to test for savings during relearning (when the cursor rotation was introduced a second time) and to determine if all groups showed comparable savings.

## Results

### Baseline motor control

Before investigating how age affects visuomotor adaptation processes, we first explored how age affects baseline motor performance (Fig 2). We found no significant differences in reach accuracy across age-groups (Fig 2A), with moderate support for the null hypothesis ($F_{(4, 126)}$ = 1.792, p = 0.134, $\eta^2$ = 0.054, $BF_{10}$ = 0.293). However, we found a significant effect of group on reach variability ($F_{(4, 126)}$ = 6.735, p < 0.001, $\eta^2$ = 0.176, $BF_{10}$ > 100), and extreme evidence in

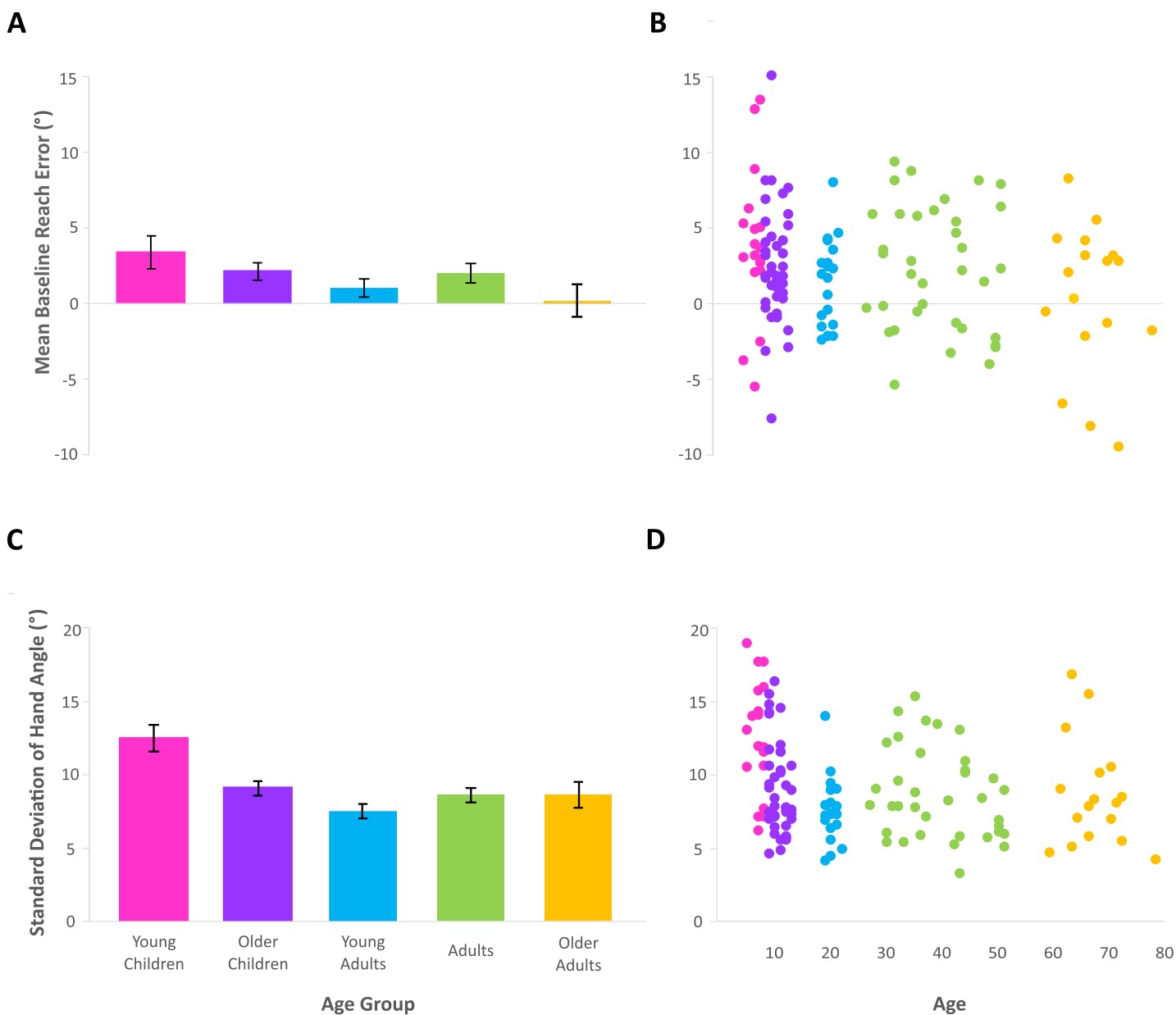

**Fig 2. Baseline motor performance.** Means are taken from the last 9 trials of baseline and error bars represent standard error of the mean. (A) Baseline accuracy is no different across age groups. (B) Individual data points for baseline accuracy. (C) Young children exhibit greater variability, compared to other age groups, during baseline reaches. (D) Individual data points for baseline variability.

support of the alternative hypothesis (Fig 2C). Tukey post-hoc tests revealed that this difference was only between the young children and all other groups. We tested differences in mouse experience between young children and older children, using a Welch T-test, and found that young children were indeed less experienced using a mouse ($t_{(29.45)} = 2.441$, $p = 0.021$, $d = 0.715$, $BF_{10} = 3.317$). There was also a significant correlation between reach variability and mouse experience in children ($p = 0.005$, $r = -0.376$, $BF_{10} = 8.889$). Taken together, these findings suggest that motor accuracy remains stable as we age, but that young children

are almost 50% more variable in their baseline motor performance when operating a mouse, which could be related to their overall experience with the device.

### Learning rate

Fig 3A shows the reaching performance across all trials for the five groups, while Fig 3B shows boxplots for the blocks of trials we used for analyses across the last 3 phases of the experiment. When investigating the learning rate during initial training (training blocks 1, 2, 5 and 15 in Fig 3B), as expected we found a statistically significant effect of block reflecting the usual learning pattern (F (2.77, 348.93) = 490.74, p < .001, $^2$ = 0.610, $BF_{10} > 100$), where performance gradually improves over the course of training. More importantly, we found no clear effect of age across the five age groups (F (4, 126) = 2.15, p = 0.078, $^2$ = 0.014, $BF_{10} = 0.559$) and only a small, but significant, 5 X 4 interaction between group and block (F (11.08, 348.93) = 1.877, p = 0.041, $^2$ = 0.009, $BF_{10} = 0.423$) which, in both cases, only suggests anecdotal support for the null hypothesis. For completeness, we conducted a Bonferroni post-hoc analysis on the significant interaction, which revealed that, although there was a significant difference between block 1 and block 2 of learning for the young children, there was no further significant differences found in this group between block 2, block 5 and block 15 of learning. In contrast, significant differences were found between all these blocks for all other age groups (training blocks in Fig 3B). Taken together, this suggests that young children might be reaching asymptotic levels slightly faster than older age groups.

### Aftereffects and unlearning

Next, we confirmed that all groups produced significant reach aftereffects (p < 0.001, $BF_{10} >$ 100). More importantly, we found that these aftereffects in block 1 were not significantly different across the age groups (F (4, 126) = 1.087, p = 0.366, $^2$ = 0.033, $BF_{10} = 0.120$), with moderate support for the null hypothesis (washout block 1 in Fig 3B). Not surprisingly, especially given the cursor was visible, the reach deviations decreased across the washout phase (F (1.95, 245.15) = 50.586, p < .001, $^2$ = 0.153, $BF_{10} > 100$). Again, this rate of decay in adaptation, or de-adaptation, did not differ across age groups, as there was no significant interaction between group and block (F (7.78, 245.15) = 0.838, p = 0.567, $^2$ = 0.010, $BF_{10} = 0.030$) with very strong support for this null hypothesis. Taken together, this suggests that all groups produced similar aftereffects which de-adapted at a similar rate.

As we can see in Fig 3B, it appears that none of the groups returned to baseline performance by the end of the 15 washout trials (washout block 5 in Fig 3B). If that were the case, then baseline-corrected reach errors in this block should have been close to 0. Indeed, each group did show persistent aftereffects during this last block of washout (for all groups, p = 0.008, $BF_{10} =$ 7) but the deviation again did not differ across age-groups (F (4, 126) = 2.146, p = 0.079, $^2 =$ 0.064, $BF_{10} = 0.517$). Although Bayesian analysis only suggests anecdotal support for the groups being similar in this block, the purpose of these tests was just to confirm that none of the groups returned to baseline performance by the end of the washout phase; aftereffects were still present for all groups when they entered the relearning phase.

### Generalization of learning

We also used performance during the washout phases to measure generalization. Fig 3C shows the learning performance across all targets, collapsed across all trials, for the five groups during the washout phase. When investigating the amount of adaptation at different distances from the trained target (0˚ was the distance from the trained target, while 45˚ and 90˚ are the distances untrained targets were from the trained) we found a statistically significant effect of

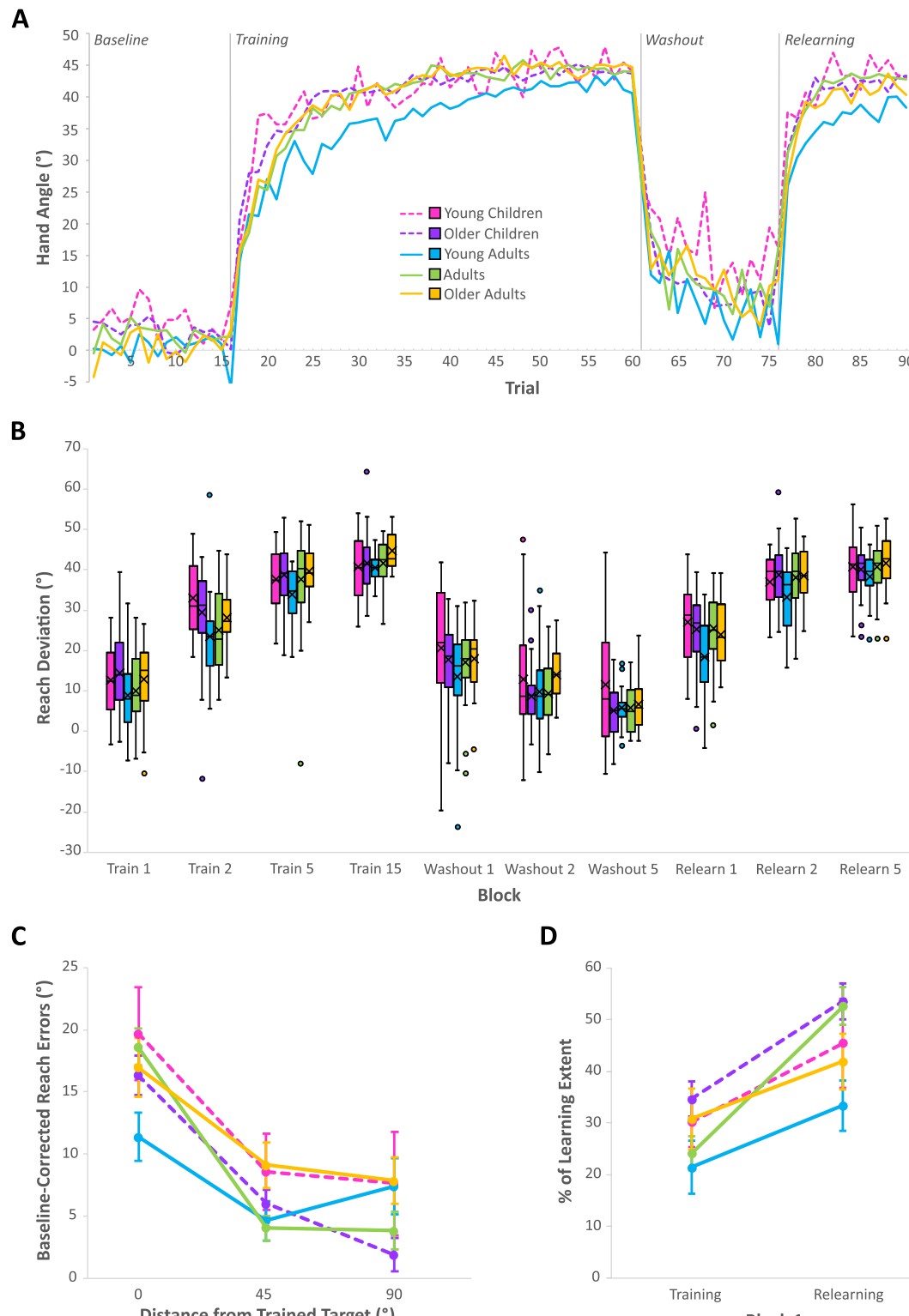

**Fig 3. Visuomotor learning performance across age groups.** (A) Performance across trials. Solid and dashed lines represent group means. (B) Baseline-corrected group performance across blocks, and phases, used for analysis, shown with boxplots. The boxes represent the interquartile range, with the "X" representing the mean and the middle line representing the median. Outliers are shown with dots. (C) Generalization of learning to different distances of untrained targets (45° and 90°) targets, from the trained target (0° difference) during the washout phase. (D) Savings during relearning. (C & D) Group means are shown by dots and error bars represent standard error of the mean.

target distance, reflecting the usual generalization pattern where there is limited transfer of learning to untrained target locations (F (1.18, 228.55) = 56.03, p < .001, $^2$ = 0.184, $BF_{10}$ > 100). Bonferroni post-hoc analysis confirmed that these differences were between the trained target and both untrained targets but that learning at the untrained targets was no different from one another. More importantly, we found this generalization pattern did not differ across age-groups, as there was no significant interaction between group and block (F (7.27, 228.55) = 1.99, p = 0.055, $^2$ = 0.026, $BF_{10}$ = 0.762) and anecdotal support for the null hypothesis. Taken together, this suggests that generalization patterns are consistent across the lifespan; we can see in Fig 3C that all age groups show limited generalization to untrained target locations through-out the washout phase.

### Relearning rate and savings

Then, we wanted to explore group differences in the rate of re-learning during the last phase of the experiment. When investigating the relearning rate (re-learning blocks 1, 2 and 5 in Fig 3B), similar to our training analysis, we found no effect of age group, with only anecdotal support for the null hypothesis (F (4, 126) = 2.109, p = 0.083, $^2$ = 0.027, $BF_{10}$ = 0.722), nor a significant interaction between group and block, with moderate support for the null hypothesis (F (7.86, 247.66) = 1.658, p = 0.111, $^2$ = 0.008, $BF_{10}$ = 0.173), but just the expected effect of block (F (1.97, 247.66) = 353.393, p < .001, $^2$ = 0.413, $BF_{10}$ > 100). Bayesian analysis suggested this difference was extreme and Bonferroni post-hoc tests revealed that there were significant differences between all three of the blocks. Taken together, this suggests that all groups re-learned to counter the cursor rotation at a similar rate.

Lastly, we compared the percentage of learning extent attained in block 1 of training to that attained in block 1 of relearning, using the re-corrected reach deviations for the relearning phase described earlier, and tested for possible group differences in savings (Fig 3D). Here we found a significant effect of phase (F (1, 126) = 36.64, p < 0.001, $^2$ = 0.105, $BF_{10}$ > 100), which suggests savings, or faster-relearning, but, more importantly, no significant interaction between phase and group (F (4, 126) = 1.50, p = 0.205, $^2$ = 0.017, $BF_{10}$ = 0.329) further suggesting that savings was largely unaffected by age-group. Although not expected, we also found a significant, but weak (anecdotal evidence for), effect of group (F (4, 126) = 3.15, p = 0.017, $^2$ = 0.047, $BF_{10}$ = 1.550) in this analysis. Results of a Bonferroni post-hoc test found that the group difference was only due to differences between the older children (purple/grey dashed lines in Fig 3D) and young adults (blue/black solid lines in Fig 3D), with young adults compensating a bit less for the perturbation than older children. We noticed that young adults in this study seemed to show slightly poorer performance than other groups throughout most phases of the study (Fig 3A and 3B), although not enough to be detected until this comparison. This group difference, for which Bayesian analyses only suggested anecdotal support for, merely highlights slight differences in learning extent across both blocks; since we did not find a significant interaction, with moderate support for the null hypothesis, we can conclude that all groups showed similar patterns of savings.

### Discussion

The main goal of this study was to compare visuomotor learning performance, for various properties of learning, across the lifespan. Specifically, we quantified the rate, and extent, of visuomotor adaptation when a 45˚ CCW cursor was abruptly introduced, as well as during washout, generalization, and rate of re-learning. Prior to comparing adaptation to this visuo-motor distortion, we first confirmed that baseline accuracy, when reaching with an aligned cursor, was similar across the age groups. Only young children showed significantly more

variability operating the mouse relative to the other age groups. All groups adapted quickly, and almost completely, in both phases where they were exposed to a 45˚ rotation in this short, gamified learning task. These learning curves closely overlapped across the age groups (Fig 3A and 3B) with no evidence for differences in learning extent. At most, we found some weak evidence that young children seemed to adapt their reaches at a slightly faster rate. All groups continued to deviate their cursor even when the 45˚ cursor-rotation was removed; that is, they showed significant aftereffects. These aftereffects, and subsequent rates of de-adaptation, were similar across age groups. As expected, aftereffects were present not only for the trained target, but also to a smaller extent for the untrained targets. Yet, this typical pattern of generalization did not differ as a function of age group. All groups relearned to counter the cursor rotation at a similar rate, and this was significantly faster than the first time participants were exposed to the rotation. That is, all groups showed similar savings. In summary, our results suggest that visuomotor learning performance, across multiple characteristics, does not differ across the lifespan.

## Baseline motor control

Our finding that baseline accuracy is no different across the lifespan for simple cursor-to-target movements (Fig 2A and 2B) is supported by other literature. Studies that compare groups of children of various ages have found no significant differences in baseline accuracy when reaching to visual targets with an aligned cursor [7, 8]. The same is true when a group of young adults are included for comparison with groups of children [8, 9]. Additionally, baseline accuracy of older adults is no different than younger adults [14, 16, 18, 21, 26]. Thus, general visuomotor accuracy in this kind of simple cursor-to-target task remains stable across the lifespan and cannot explain any of the differences found between groups in the studies we review below.

In our study, we found that young children were significantly more variable (by almost 50% more than other age groups) in their baseline reaching performance (Fig 2C and 2D), which is also supported by other work. In a previous study, young children (aged 5–6) were found to be almost 30% more variable than older children (aged 7–8 and 9–10) at baseline [8]. Ferrel et al., (2001) also found reaching performance of 6-year-olds to be almost 30% more variable than older children or young adults, despite finding no differences in baseline accuracy (which would be independent of variability, like in our study) and has been attributed to less fine-tuned motor control [7]. It is possible that this also explains our results, since we found that children with less experience using a mouse tended to be more variable in their baseline motor control. However, in this simple task, we, along with others [26, 27], found that older adults are no more variable than younger adults during these aligned-cursor reaches. Overall, the results suggest that variability in reaches made with an aligned cursor is consistent across age, apart from young children.

## Extent and rate of adaptation

Although we found young children were more variable in their baseline reaching performance, this did not impact their rate, or extent, of visuomotor adaptation as we found no major differences between any of our groups for either aspect (Training Blocks in Fig 3B). Our study was designed to be short and engaging to ensure our young participants were motivated to do well and not become fatigued, or bored. Past work has indicated that children younger than 8 may have difficulties adapting their reaching movements to a rotated cursor, especially for rotations larger than 90˚, but that 11-year-olds perform similarly to adults for rotations of all sizes, albeit with slower movement times [9]. Given that their participants only completed 8 trials for each

of the 5 rotations tested, this may not have been enough trials to sufficiently capture children's visuomotor learning abilities. However, a study that imposed a 45˚ rotation (the same as in our current study), over 60 training trials, still found that 4-year-olds had some trouble adjusting their movements, as their initial movement plans suggested less compensation for the rotation compared to older children (6 or 8-years-old). They found that 8-year-old children compensated for it more than both younger groups, although all groups of children were able to increase their compensation by the end of the adaptation phase [7]. Likewise, using a slightly larger (60˚) rotation and over 100 training trials, King et al., (2009) found younger children compensated about 25–30% less than older children, although both groups doubled their amount of compensation in the last training block, compared to the first. The only study that truly measured children's rate of adaptation, to a 45˚ rotation, found that rate of early adaptation, but not late adaptation, was slower in children. However, this study did not have gamified stimuli and required participants to reach to 8 target locations during training (we only had 1). This study is also the only other one we are aware of to compare children to both older adults (discussed below) and younger adults, like we did [11]. Taken together, children can adapt to visuomotor rotations, although the youngest children may adapt less well than slightly older children, especially when visuomotor adaptation tasks are more challenging, or less fun. These results could be explained by the OPTIMAL theory of motor learning, which argues that motivational and attentional factors strengthen goal-action pairings, ultimately leading to improved motor performance [28].

A major difference between our results and those of previous studies involving children is how quickly, and the overall extent of which, children (and even adult controls) compensated for the 45˚ cursor rotation. All groups compensated for the rotation by at least 90% by the end of our 45 training trials (Training Block 15 in Fig 3B) and if we look at their performance across all trials (Fig 3A) it appears that this large compensation occurred early (within the first 25 trials). In most previous studies, nearly all of which had more training trials, participants compensated for the rotation at most by 80%. For example, Kagerer and Clark (2014) found that both adult controls and children only achieved 70% compensation of a 60˚ visuomotor rotation. This suggests that some of the poor performance in children (and perhaps adults) may be partly due to lack of motivation and fatigue.

Given the protocol and lab setting of previous studies, it is possible that children were not fully attending to the task, since it only involved moving one dot to another. Therefore, we shortened and gamified our visuomotor learning task to make it more engaging for children. Anticipating seeing new aliens, and planets, likely enhanced children's external focus on the task goal, which satisfies the attentional component of the OPTIMAL theory [28]. The children in our study also used a child-size mouse rather than a digitizing pen, perhaps leading to more confidence, or self-efficacy, in completing the task. Lastly, given that our task was portable, we ran children in a fun, familiar environment for them (at a library or camp), which likely supported positive affect, which is also thought to promote self-efficacy, a crucial motivational component in the OPTIMAL theory [28]. All these aspects should have made the entire experience less intimidating and less boring; this could be more motivating for children to learn and might explain the greater compensation that we observed early in training for all our age groups. The only group who showed any departure from this excellent performance (although non-significantly so) were our young adults, who were the only group that would have done this study in a lab setting, and for course credit, rather than purely out of interest. Our results suggest attention and motivation may play an important role in motor learning performance, especially for special groups like children.

The effect of aging on visuomotor adaptation is somewhat mixed for older adults. Generally, there is more consistent evidence that for larger rotations, i.e. 60˚ or larger, older adults

adapt at a slower rate, and to a lesser extent than young adults [17–20, 29–31], which has been attributed most to aging-related deficits in the explicit or cognitive component of learning [13, 15, 22, 30, 31]. Furthermore, applying transcranial direct current stimulation (tDCS) to the cerebellum [31, 32], or M1 [30, 31], can counteract these deficits in older adults' adaptation performance. For smaller rotations, usually 30°, the impact of aging on visuomotor adaptation is less clear. Some studies find that older adults do not differ from younger adults in the rate or extent of their adaptation [13–15], but others have found that older adults adapt to a lesser extent [16, 21, 32] and at a slower rate compared to young adults [16]. Another study, using the same 45° rotation as we did, but having participants train at 8 target locations, found that older adults adapted at a slower rate during the early stages of adaptation, but not during the later phases, which, like studies involving even larger rotations, was attributed to impaired explicit learning processes [11]. As mentioned previously, this is the only other study we are aware of that has compared performance of older adults to both children and younger adults within the same paradigm. Again, we found no difference in adaptation performance across age for a 45° rotation, which may suggest that in our short, gamified study, older adults did not find this moderate rotation size exceptionally challenging compared to younger adults, especially when they only had to train at a single target location. The shortness of the study also meant we didn't include additional tasks, or instructions, to be able to distinguish the extent by which implicit and explicit components contribute to the adaptation performance; future research should explore this issue. Given that we found no differences in adaptation performance between our groups, particularly in the early training blocks, it is possible that we have found no evidence of impaired explicit learning, but this was not a focus of the current study. Since all our age groups adapted at the same rate, and to the same extent, we then investigated whether age affected other aspects of adaptation, such as aftereffects, de-adaptation, generalization, and savings.

## Aftereffects, rate of unlearning and generalization

Following adaptation, our participants continued to deviate their mouse movements during washout, even though the hand-cursor was realigned once more; that is, they displayed significant aftereffects (Washout Block 1 in Fig 3B). More interestingly, the magnitude of these reach aftereffects did not differ across the age groups. Although the aligned cursor was visible in our study, the reach aftereffects for the first block of 3 trials (to the trained and untrained targets) were about 17°, which is typically the size of reach aftereffects elicited when the cursor is not visible in young adults [2]. Thus, our gamified study, with only a short bout of training, was able to elicit robust reach aftereffects across the lifespan.

One of the first studies to measure aftereffects in children, after adapting to a 45° rotation, found that the younger children (aged 4 and 6) did not produce these typical aftereffects, while 8-year-olds did [7]. However, their results are possibly confounded since these two youngest groups also did not fully compensate for the 45° cursor rotation during training, compared to 8-year-olds. Subsequent studies by King et al. (2009), and Kagerer & Clark (2014), have found that 5–6-year-old children showed a similar magnitude of reach aftereffects as older children and adults. Thus, consistent with our findings, young children, who can adapt to a visuomotor rotation, produce aftereffects that are comparable to older children and adults.

Not only did we investigate the initial trials during washout to gauge for reach aftereffects, but we also compared de-adaptation rates across the age groups. We found that rate of de-adaptation to the cursor rotation did not differ across the age groups (Washout Block in Fig 3B). Unsurprisingly, given that we only had 15 trials of washout, de-adaptation was not complete at the end, like in the study by Kagerer and Clark (2014), although they included 35

trials, and the cursor was not visible during these washout trials. Yet, Kagerer and Clark (2014) found that both the rate, and extent, of de-adaptation in young children (5–6) was smaller than that of older children (11–12), as well as young adults. This discrepancy, between their results and ours, could be due to different learning processes when the aligned cursor is visible, compared to invisible, during washout. Typically, aftereffects will decay faster when the hand-cursor is still visible, due to concurrent de-adaptation, compared to if the cursor is removed entirely during aftereffect trials [3]; having the cursor visible in our study may have helped younger children de-adapt quickly. However, the availability of visual feedback is unlikely to be the full explanation since a prism study by Gómez-Moya et al. (2016) found a significant interaction between age and trial number during de-adaptation [33]. To test for aftereffects after prism adaptation, the prisms are removed during de-adaptation and, therefore, aligned visual feedback is available because they can see their own hand. Although they did not run post-hoc tests, it does seem that their significant interaction was due to slower rates of unlearning in the two youngest groups (4–5 and 7–8) compared to adults. This was despite finding no significant effect of age on rate, or extent, of adaptation to the displacing prism. Furthermore, there were no age differences in reach aftereffects captured during the first trial of their washout phase, yet they still found children de-adapted slower than adults even though this aligned feedback was present. Ruitenberg et al. (2023), who, like us, used a 45˚ rotation and had a visible cursor during washout also found that children de-adapted at a slower rate than adults, however, they also found slower rates of early adaptation in children and had more training targets than we did here. Therefore, whether young children de-adapt slower, and whether this depends on the availability of visual feedback during training, or if it is dependent on the number of trained locations, requires further investigation. In our study, to better engage all participants, we opted to make the cursor visible during washout, rather than invisible, since reaching with an invisible mouse cursor would have made the study much longer, and far less enjoyable, especially for the children. Thus, we can only conclude that for short bouts of gamified visuomotor learning, adaptation and de-adaptation may be equivalent across the lifespan.

Our finding that older adults show comparable aftereffects, as well as similar rates of unlearning, as younger participants is much more in line with previous work, even when comparing our results to studies where older adults have showed reduced compensation for the rotation during the training phase [19, 21]. Regardless of the size of the cursor rotation used during training, so long as participants were not cued, or instructed, to evoke a cognitive strategy, older adults have consistently been found to show significant aftereffects that are of a similar magnitude as young adults [13, 15, 26] which decay at a similar rate [14, 21, 30–32]. It is only when explicit strategies are evoked, after adaptation to cursor rotations larger than 45˚, that older adults produce smaller aftereffects than young adults [13]. When the rotation is smaller, even when eliciting a cognitive strategy during aftereffect trials, older adults again show aftereffects that are a similar size as young adults [13, 15]. Since aftereffects are often considered a measure of implicit learning [1], it is possible that preserved implicit processes in older adults can counteract deficits in explicit learning during aftereffect trials [22], at least after adapting to smaller rotations. Although our study cannot distinguish between implicit and explicit processes, it is unlikely that explicit strategy contributed that much to either the adaptation or de-adaptation given the rotation size was not so large. Thus, our results are generally in line with most of the past recent literature showing comparable rates and extent of adaptation/de-adaptation for small visuomotor rotation for younger and older adults.

Finding comparable levels of aftereffects, as well as similar rates of unlearning, for all groups during the washout phase permitted us to explore possible group differences in generalization patterns towards untrained target locations. Despite having only 15 washout trials in total, in addition to the cursor being visible during these trials, we found the typical generalization

pattern [4, 34, 35], whereby people showed the largest deviations at the trained target (45˚) and significant, but reduced deviations when reaching towards nearby, untrained targets (90˚ and 135˚) (Fig 3C). Specifically, we found that reaches toward both untrained locations were almost 3 times smaller, on average for all groups, than those towards the trained location, but more importantly, this generalization pattern did not differ across any of our age groups. To our knowledge, our study is the first to examine generalization patterns in children following visuomotor adaptation. We are aware of one study that has investigated generalization in older adults; Heuer & Hegele (2008) found that older adults showed comparable generalization patterns as young adults, such that after adapting to a 75˚ rotation at a single trained target location, and having them reach with an invisible cursor during washout trials, aftereffects at nearby target locations were about half the size of those at the trained location, like in our study (where we provided vision of the realigned cursor). Overall, our results suggest that our ability to generalize visuomotor learning to new, nearby locations is present in early childhood and remains stable throughout the l.

## Relearning rate and savings

Finally, we found that all groups relearned to counter the cursor rotation at a similar rate, and that all groups showed evidence of savings, when the rotation was introduced a second time (Fig 3D). Although we did find an effect of group, we attribute this minor difference in performance of young adults to their motivation for participating, as discussed above. Since we found a significant effect of phase (training vs relearning), but no significant interaction between phase and group, we can conclude that savings was similar across all age groups.

Most of the previous work on savings has focused on young adults, but there have been a few studies that have examined savings in older adults; studies testing savings in children are incredibly scarce. The only study we found that tested for savings in children found no evidence of a quicker relearning rate, to the 60˚ cursor rotation they first trained with, in older children (aged 10–12) after a break of at least 10 hours [12]. However, this study did not include a group of young adults for comparison, making it difficult to know if their null result was really an effect of age. Since Kitago et al. (2013) found that young adults produce greater savings after experiencing a washout phase, in which the cursor is visible but realigned with the hand's position, compared to unlearning over time alone (general forgetting), it is possible that the discrepancy between our results and those of Urbain et al. (2014) are simply due to differences in paradigm; 15 washout trials may not be an equivalent break in time compared to a 10-hour delay. The few studies which test savings in older adults also showed older adults exhibit similar savings as young adults, like we did, but slightly lower adaptation overall [22, 23], while we found no differences in initial adaptation. The rarity of these studies, in addition to the discrepancies between their findings and ours, warrants future replication of our results. Thus, in the context of short, gamified situations, our study suggests that both adaptation and savings is no different across the lifespan.

## Limitations

While only having participants adapt to a single target during training allowed us to have a shorter experiment, and to measure generalization, in addition to many other features of motor learning, this may also explain why we did not find an effect of aging; many other studies involving children, or older adults, have included 4–8 targets during training, which could explain why they have found impairments, and may better reflect real-world adaptive processes. While we agree that having more target locations might tap into impaired cognitive/strategic abilities in these age groups, this was not a focus of the current study. However, since

we are one of the first to measure several aspects of learning across multiple age groups, our results should still be relevant to researchers in various fields.

## Conclusion

In this study we found that visuomotor learning abilities are similar across the lifespan; we found no differences in performance between young children, older children, young adults, adults and older adults for adaptation extent, adaptation rate, aftereffects, de-adaptation rate, generalization, or savings. Our results differ from past studies, which sometimes, but not always, find an effect of age on visuomotor adaptation. Past findings are also confounded by the fact that these experiments can be rather long, boring, and may involve a special trip to an unfamiliar laboratory. This is not the case for young adults who tend to be students at the university where the research is being conducted. By testing children and older adults in a more familiar setting, such as at a library or camp, we found that visuomotor adaptation performance was similar across the lifespan, at least for short bouts of gamified learning.

## Supporting information

**S1 File. Learning rates.**
(PDF)

## Acknowledgments

Special thanks to the Innisfil Public Library and the York Science Camps for helping us promoting the study to parents and providing spaces for us to conduct our experiment. Thank you to undergraduate students Aqib Mannan and Safiya Erdogan for assisting with data collection.

## Author Contributions

**Conceptualization:** Holly A. Clayton, Bernard Marius 't Hart, Denise Y. P. Henriques.

**Data curation:** Bernard Marius 't Hart.

**Formal analysis:** Holly A. Clayton.

**Funding acquisition:** Denise Y. P. Henriques.

**Investigation:** Holly A. Clayton, Sahir Abbas.

**Methodology:** Bernard Marius 't Hart, Denise Y. P. Henriques.

**Project administration:** Holly A. Clayton, Denise Y. P. Henriques.

**Resources:** Denise Y. P. Henriques.

**Software:** Bernard Marius 't Hart.

**Supervision:** Denise Y. P. Henriques.

**Visualization:** Holly A. Clayton.

**Writing – original draft:** Holly A. Clayton, Sahir Abbas.

**Writing – review & editing:** Holly A. Clayton, Sahir Abbas, Bernard Marius 't Hart, Denise Y. P. Henriques.

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
