## [Decision Letter · Decision Letter 0]

9 Nov 2023

PONE-D-23-26875Visuomotor adaptation across the lifespanPLOS ONE

Dear Dr. Clayton,

Thank you for submitting your manuscript to PLOS ONE. After careful consideration, we feel that it has merit but does not fully meet PLOS ONE’s publication criteria as it currently stands. Therefore, we invite you to submit a revised version of the manuscript that addresses the points raised during the review process.

We look forward to receiving your revised manuscript.

Kind regards,

Yih-Kuen Jan, PhD

Academic Editor

PLOS ONE

Reviewers' comments:

Reviewer's Responses to Questions

**Comments to the Author**

1. Is the manuscript technically sound, and do the data support the conclusions?

Reviewer #1: Yes

Reviewer #2: Yes

Reviewer #3: Yes

2. Has the statistical analysis been performed appropriately and rigorously? 

Reviewer #1: Yes

Reviewer #2: Yes

Reviewer #3: Yes

3. Have the authors made all data underlying the findings in their manuscript fully available?

Reviewer #1: Yes

Reviewer #2: Yes

Reviewer #3: Yes

4. Is the manuscript presented in an intelligible fashion and written in standard English?

Reviewer #1: Yes

Reviewer #2: Yes

Reviewer #3: Yes

5. Review Comments to the Author

Reviewer #1: In this article the authors investigate visuomotor adaptation capabilities across the lifespan (from young children to older adults) using a shortened, gamified task. They report similarities in learning, unlearning, savings, and generalisation between all age groups. Overall, the manuscript is well presented and reasoned, however, there are some concerns that need addressing before suitable for publication.

Major Comments:

1. A main concern with the manuscript lies in the choice of task used and thus the generalisability of results to the wider literature. It is understood that the shortened and gamified version of the task was chosen to keep the task brief, and participants engaged (particularly younger children), however, the decision to only assess adaptation towards one target may have had significant implications on the results, specifically age-related differences. It is likely that task constraints of only adapting to one target would be much less than adapting to four or eight targets, as described in much of the existing literature. This is particularly pertinent, as impaired cognitive/strategic abilities have been attributed to reduced adaptive performance in younger children and older adults. While the benefits of the adapted task are discussed throughout, the potential caveat to the results needs to be addressed in the discussion or a limitations section, especially as real-world adaptive processes involve complex interactions between multiple sensory inputs and motor outputs.

2. The sample sizes for each group differ significantly in many of the age-groups. This is not acknowledged or discussed in the manuscript and no power calculations are included. The authors should at least provide an explanation or rationale for this. They should also report whether the data is normally distributed and adjust analysis appropriately e.g., consider using median scores rather than means.

3. Some aspects of the ‘Data Analysis’ section are written passively, with ‘then’ and ‘next’ repeated multiple times in each subsection making it difficult to follow exactly why each test was run and for what reason. This should be re-drafted throughout, for example, instead of “Then we wanted to confirm that each group showed aftereffects by performing 5 separate single-sample t-tests, for each of the groups, to see if their reach deviation for the first block of washout was significantly different from 0.”, consider changing to something like “To confirm that each group showed aftereffects, we performed 5 separate one-sample t-tests (one for each group), determining whether reach deviation during the first washout block was significantly different from 0.” Etc.

4. Figure 2 should be edited to include individual data points and/or the authors should consider re-plotting as box plots (similar to Figure 3) to improve interpretation of these results.

5. The authors nicely highlight the differences in variability between young children and the other age groups during baseline trials in the discussion. However, they do not attempt to explain this result with evidence from the wider developmental motor control literature. Some reasoning on this matter should be included.

6. Following on from the previous point, the authors do not carry out (or have not presented results) on variability for the other phases of the task. It seems strange to only conduct this analysis for baseline trials (unless there is a specific reason – which should then be included in the methods). It appears like younger adults are much more variable than the other age-groups during washout trials as well, which may be an interesting point to investigate further. At the very least, a rationale for only looking at variability during baseline reaches needs to be included.

Minor Comments:

1. Line 49: The authors state that “young adults usually adapt quickly and completely”. However, this is not necessarily true. In fact, there is whole body of literature dedicated to understanding why individuals are unable to fully compensate for visuomotor rotations. Please alter the sentence so it reflects the literature.

2. Ruitenberg et al., 2023. Developmental and age differences in visuomotor adaptation across the lifespan. This relatively new paper aims to answer similar questions but finds contrasting results and should be introduced and discussed in the manuscript.

3. Line 65-67: This sentence feels out of place at this point in the introduction “Most studies focus on the overall magnitude of adaptation, rather than its speed, so we compared changes across blocks of 3 trials to better capture potential group differences in learning rate.” The authors should consider removing it.

4. The authors tend to underrepresent the wealth of evidence suggesting visuomotor adaptation is impaired in older adults – maintaining the line that evidence is relatively balanced with respect to adaptive capabilities. This tone should be altered, and key references should be added: Hardwick & Celnik, 2014; Panouillères et al., 2015; Weightman et al., 2020 (tDCS papers reporting age-related visuomotor impairments in the respective young and older adults sham stimulation groups, with different rotation magnitudes), & Vandervoorde & Orban de Xivry, 2019, J Physiology.

5. Please provide exact numbers of right- and left-handed participants and make it clearer that all participants used their right-hand regardless of handedness.

6. Why were the last 9 trials of baseline used for analysis – was there any specific rationale for this?

7. Figure 3: The authors state “solid lines represent group means”, however there are also dashed lines representing the means of the young and older children’s groups. This needs correcting to avoid confusion.

8. Figure 3: The authors should report what each aspect of the box plots represents e.g., boxes indicatie the mean, upper and lower quartiles and the range is shown via error bars.

9. A few times in the manuscript the authors write “the usual learning pattern” (Line 286/7) or “the usual generalization pattern” (Line 331), this should be edited to be more specific.

10. Line 385: “Not only did all groups also relearn to counter the cursor rotation at a similar rate, but this re-learning was significantly faster the second time participants were exposed to the rotation (known as savings) and, again, was independent of age”. Please remove the final part of this sentence as it repeats the info at the beginning (results were not different across age-groups), or re-draft as desired.

Reviewer #2: In this paper the authors investigate visuomotor learning abilities throughout the lifespan, by comparing rate of adaptation, extent of learning, rate of unlearning, generalization, and savings, in cohorts of young children, older children, young adults, middle-aged adults, and older adults. They find similar results across all ages groups.

This paper looks to resolve inconsistencies in the literature regarding adaptive capacities across ageing. These inconsistencies likely stem from heterogeneity in paradigms that have been used in the past. The authors opt to simultaneously compare several different tasks that probe adaptation across multiple age groups. This is a daunting task, as it is not at all clear whether all these tasks should be similarly susceptible to ageing, as we still have an incomplete understanding of the extent to which their mechanistic bases are common. In fact, it is not clear whether the hypothesis is that all tasks should necessarily show a similar pattern across ageing or not. In this light, I’m not convinced that this strategy represents a fruitful way to “settle” the issue. The fact that there is no effect of ageing certainly adds to the literature, but is inconsistent with many papers that have reported age-related differences, so the impact of the finding is unclear. How does the present results reconcile the previous findings?

Here are some comments to enhance the paper:

Abstract: please add the mean age of the different groups

Introduction: p5 line 84 “Hegele”

Methods: The number of ppts per group is quite different. Can you confirm that this did not influence the statistical results?

There is a relatively low number of datapoints for many measurements (eg. 9 trials for baseline motor control). Under the presence of variability, this may have limited capacity to detect differences and lead to the null effects. The authors should clearly highlight this limitation.

Discussion: I have a hard time with the interpretation that baseline reaching accuracy is similar across age groups if variability is greater by 50% in children. I get it from the way the dependent variables are calculated here, but taken more broadly, variability would be expected to influence accuracy.

The authors do not provide much mechanistic explanation to interpret their findings. P21 line 428 provides one such opportunity: how would fun be related to adaptation? If the authors bring this up, do they have literature to back it up? P22 line 451 provides another: how would task quality, and environment, be related to adaptation? Overall this paper would benefit from being less observational/descriptive, and more mechanistically-inspired, hypothesis-driven.

P25 line 529: It is not clear why the authors would not expect explicit processes in their task, as it has been well documented for such perturbation magnitude (45deg).

Reviewer #3: This manuscript describes a study looking at visuomotor learning across the lifespan. It groups participants into young children, older children, young adults, middle-aged adults, and older adults. Participants had to adapt to a visuomotor rotation during a gamified computer/mouse task. The results indicate that age does not impact rate of adaptation, rate of unlearning, generalization, or same-day re-learning. The manuscript is well written and adds important insight into visuomotor learning (especially by including children). With that said, I have several comments that should be addressed.

1. What is the rationale for the specific age ranges to separate age groups? In addition, why are late 20-year-olds part of the middle age group? Typically, middle age is between 40 and 60 (give or take). The choice of age ranges requires justification.

2. The authors quantified mouse use among the children. What about older adults, especially those over 70? Many seniors are not computer savvy.

3. Line 202: why the last 9, and not 10?

4. For the mixed ANOVAs, how were repeated measures within-participants handled? In JASP, was the repeated measures ANOVA function used? It isn’t clear.

5. Aftereffects: given that visual feedback is available during the washout block, one would expect changes driven by this feedback. The authors do discuss this in a few places in the discussion section. However, for testing aftereffects, why not use the first trial only? This would be the “true” aftereffect. An average of the first 3 trials seems inappropriate for that aspect of the analysis.

6. Lines 242-247: Can this really be called a rate?

7. Some of the Bayes Factors are between 0.33 and 3, and thus anecdotal. However, values close to 1 might suggest the sample size of the groups is too small to detect differences. Have the authors considered this?

8. Given the large number of participants overall, the authors should consider running correlations between age and different learning performance measures. Age group might not have an effect, but these correlations might suggest age does a play a role (if significant). I believe this would be a worthwhile analysis, regardless of the outcome.

6. PLOS authors have the option to publish the peer review history of their article (what does this mean?). If published, this will include your full peer review and any attached files.

Reviewer #1: No

Reviewer #2: No

Reviewer #3: No

---

## [Author Response · Author response to Decision Letter 0]

9 May 2024

We have included our responses in a separate document that we attached.

---

## [Decision Letter · Decision Letter 1]

14 Jun 2024

Visuomotor adaptation across the lifespan

PONE-D-23-26875R1

Dear Dr. Clayton,

We’re pleased to inform you that your manuscript has been judged scientifically suitable for publication and will be formally accepted for publication once it meets all outstanding technical requirements.

Kind regards,

Yih-Kuen Jan, PhD

Academic Editor

PLOS ONE

Additional Editor Comments (optional):

Reviewers' comments:

Reviewer's Responses to Questions

**Comments to the Author**

1. If the authors have adequately addressed your comments raised in a previous round of review and you feel that this manuscript is now acceptable for publication, you may indicate that here to bypass the “Comments to the Author” section, enter your conflict of interest statement in the “Confidential to Editor” section, and submit your "Accept" recommendation.

Reviewer #1: All comments have been addressed

Reviewer #3: All comments have been addressed

2. Is the manuscript technically sound, and do the data support the conclusions?

Reviewer #1: Yes

Reviewer #3: Yes

3. Has the statistical analysis been performed appropriately and rigorously? 

Reviewer #1: Yes

Reviewer #3: Yes

4. Have the authors made all data underlying the findings in their manuscript fully available?

Reviewer #1: Yes

Reviewer #3: Yes

5. Is the manuscript presented in an intelligible fashion and written in standard English?

Reviewer #1: Yes

Reviewer #3: Yes

6. Review Comments to the Author

Reviewer #1: The authors have addressed all raised concerns and I am happy to recommend the manuscript for publication.

Reviewer #3: (No Response)

7. PLOS authors have the option to publish the peer review history of their article (what does this mean?). If published, this will include your full peer review and any attached files.

Reviewer #1: No

Reviewer #3: No

---

## [Editor Report · Acceptance letter]

1 Jul 2024

PONE-D-23-26875R1 

PLOS ONE

Dear Dr. Clayton, 

I'm pleased to inform you that your manuscript has been deemed suitable for publication in PLOS ONE. Congratulations! Your manuscript is now being handed over to our production team.

Kind regards, 

on behalf of

Dr. Yih-Kuen Jan 

Academic Editor

PLOS ONE